# Emotion Evaluation and Response Slowing in a Non-Human Primate: New Directions for Cognitive Bias Measures of Animal Emotion?

**DOI:** 10.3390/bs6010002

**Published:** 2016-01-11

**Authors:** Emily J. Bethell, Amanda Holmes, Ann MacLarnon, Stuart Semple

**Affiliations:** 1Research Centre in Brain and Behaviour, School of Natural Sciences and Psychology, James Parsons Building, Liverpool John Moores University, Byrom Street, Liverpool L3 3AF, UK; 2Centre for Research in Cognition, Emotion and Interaction, University of Roehampton, London SW15 4JD, UK; a.holmes@roehampton.ac.uk; 3Centre for Research in Evolutionary, Social and Interdisciplinary Anthropology, University of Roehampton, London SW15 4JD, UK; a.maclarnon@roehampton.ac.uk (A.M.); s.semple@roehampton.ac.uk (S.S.)

**Keywords:** animal welfare, appraisal theory, attention bias, cognitive bias, emotion evaluation, emotional stroop, freeze, primate, response slowing, rhesus macaque

## Abstract

The cognitive bias model of animal welfare assessment is informed by studies with humans demonstrating that the interaction between emotion and cognition can be detected using laboratory tasks. A limitation of cognitive bias tasks is the amount of training required by animals prior to testing. A potential solution is to use biologically relevant stimuli that trigger innate emotional responses. Here; we develop a new method to assess emotion in rhesus macaques; informed by paradigms used with humans: emotional Stroop; visual cueing and; in particular; response slowing. In humans; performance on a simple cognitive task can become impaired when emotional distractor content is displayed. Importantly; responses become slower in anxious individuals in the presence of mild threat; a pattern not seen in non-anxious individuals; who are able to effectively process and disengage from the distractor. Here; we present a proof-of-concept study; demonstrating that rhesus macaques show slowing of responses in a simple touch-screen task when emotional content is introduced; but only when they had recently experienced a presumably stressful veterinary inspection. Our results indicate the presence of a subtle “cognitive freeze” response; the measurement of which may provide a means of identifying negative shifts in emotion in animals.

## 1. Introduction

In the last decade, there has been a notable increase in the number of studies adapting principles and methods from human cognitive psychology in order to assess animal welfare [1,2]. The most widely used of these in recent times is the “judgement bias” task [1]. In this type of task, animals must learn to discriminate abstract cues that signal reward and non-reward, before being tested on whether they make optimistic or pessimistic judgements about new abstract cues that have properties intermediate to the two learned cues. The weight of evidence from the judgement bias task supports the view that animals become more pessimistic about ambiguous information in the environment when they have undergone a negative mood manipulation or pharmacological treatment [2,3,4]. However, the initial stage of discrimination training can be time consuming (e.g., 19–43 training sessions totaling 1678–2666 trials in one study with rhesus macaques: [5]), potentially disruptive to the animals involved, lead to attrition of study subject numbers, and be costly in terms of time and money for researchers [4]. To remove the need for training on relatively complex judgement bias tasks, we have been developing complementary methods that tap into innate biases to attend to certain types of stimuli over others.

One such task is the “attention bias” task [6,7]. This requires substantially less training and has been successfully adapted for use with animals [2,8,9,10,11]. In the attention bias task, participants are presented with pairs of stimuli (e.g., one emotional and one neutral) and “attention” (e.g., eye-gaze [12,13]) to the stimuli is recorded. Evidence from humans [6] and macaques [8] shows that attention bias for emotionally-charged stimuli like facial expressions [14,15] changes with the emotion state of the viewer, suggesting attention bias provides a measure of changes in underlying affect.

While attention bias and judgement bias tasks access different emotion-cognition interactions (and therefore answer different questions about mechanisms of emotion), attention bias tasks may be more practical for measuring shifts in affective state in real world settings. Two additional lines of evidence indicate the value of further developing such tasks for assessing animal emotion. These include animal data on bodily freezing and attentional orienting to threat [16,17,18,19] and human work on a subtle cognitive form of the freeze response to threatening stimuli presented during laboratory tasks [17,20,21,22,23]. Based on these converging lines of evidence, we propose a novel approach to measure a subtle cognitive form of the freeze response to mild threat in animals, based on methods widely used with humans in the laboratory [12,20,24]. In the human laboratory studies, these tasks are considered ethically sound because they use relatively mild pictorial stimuli and tap into subtle and early processes that precede stronger negative affective states. We aim to adapt these methods to assess susceptibility to negative affect in animals by using mild pictorial stimuli that similarly tap into subtle and early processes associated with negative emotion.

The freeze response is an adaptive response to threat [17]. Following the detection of a threatening stimulus a series of innate and hard-wired defense behaviors is activated, known as the defense cascade [25,26]. Autonomic nervous and glucocorticoid stress systems produce the arousal needed for the activation of these behaviours, with freezing being an initial response that frees up resources to process the potential threat [17,27,28,29,30]. As such, freezing enhances attention and threat-processing at an early stage of the defense cascade, enabling selection of an appropriate motor response: continued freezing, flight or fight [17,31,32].

In humans, freezing in inappropriate (e.g., non-threat or low-threat) situations is used as a behavioural indicator of a dysregulated fear response [29]. People who freeze at events or situations that are only mildly threatening are considered to exhibit clinical levels of fearfulness, as seen, for example, in post-traumatic stress disorder (PTSD) [30] and social phobia [29,30]. In these clinical disorders, initially adaptive fear response (e.g., freezing) to the onset of a potential threat is maintained even if the individual subsequently perceives that no threat exists or the danger has passed [18].

In laboratory research with human and non-human animals, fear-inducing situations have been used to elicit the freeze response. The “stranger approach test” [29] and the “human intruder paradigm” [19,33] both involve high fear-eliciting contexts to test for dysregulated emotion in humans and non-human primates, respectively. For example, Buss *et al*. [29] used the “stranger approach test” to identify dysregulated emotion in children: a male stranger entered the room in which the child was playing and approached and stared at the child for up to two minutes while the child’s freezing behavior (reduction in activity >2 s) was assessed. On average, children froze for a total of ~50 s: “*freezing often involved the child appearing stuck in an unnatural or uncomfortable position*.” [29] (p. 586). Children who froze for longer had higher basal cortisol and resting cardiac activity, which were considered to indicate dysregulated fear and compromised welfare for these individuals.

Research using nonhuman primate models of human psychopathology has employed an analogue of the stranger approach test: the human intruder paradigm, HIP [33]. This test also uses bodily freezing (muscular and vocal immobility >3 s [19]) to threat as a measure of the dysregulation of fear-related behaviors. In the HIP, an intruder enters the room in which monkeys are housed and either stares at the monkey (staring is a signal of dominance and threat in macaques [34]) or stands in profile making no eye contact with the monkey (gaze aversion is a signal of subordination in macaques [34]). When stared at, monkeys typically exhibit a range of aggressive-defensive and fear behaviours, reflecting appropriate fight or flight [18]. When the stranger stands in profile, macaques tend to freeze—an adaptive response allowing assessment of the level of threat posed while reducing the likelihood of detection. Monkeys who exhibit the highest levels of freezing in the condition with no eye contact tend to have high basal cortisol levels [19] (although other studies have failed to find this relationship: [16]). These monkeys also have a tendency for increased right frontal lobe activity [16,18], which, in humans, is associated with greater negative emotion processing [35] and reactivity to negative stimuli [36]. It has been suggested that such enhanced freezing in primates reflects dysregulated emotion and fearful temperament, comparable to that seen in some human clinical disorders [18].

Animals housed in captive situations such as zoos and laboratories may encounter potential fear-eliciting situations such as approaching and staring strangers on a frequent basis. For these animals, repeated and heightened fearful response to such encounters would have deleterious effects on welfare. Animals whose emotional response to mild situations becomes dysregulated will undergo escalated levels of fear potentially resulting in pathological levels of fearfulness [17]. Animals who are able to identify the approaching stranger as mildly- or non-threatening will be quicker to process, categorise and disregard the approaching stranger, thereby effectively regulating their emotion. Tasks to identify animals with dysregulated fear response to mild threat, without the need to employ ethically problematic strong fear-eliciting contexts (physical restraint is often used: [17,37]), would be valuable.

A number of cognitive tasks have been developed for humans that use far milder stimuli and situations than those used in animal studies, that may nonetheless tap into cognitive precursor processes to fear and freezing [21]. Typically, participants are asked to perform a relatively simple repetitive cognitive task, and impairment in their performance (usually “response slowing”) is measured when emotional distractor content is introduced. The emotional distractor information may be threatening or positive, e.g., [21,24,38,39,40], depending on the study aims.

The “emotional Stroop” task is a well-known example of such a task (for a review, see [41]). It requires participants to indicate as quickly and as accurately as possible, usually using a manual key press, the colour of the ink that emotion-related words appear in [42]. These studies typically reveal a generic slowing of response when naming the colour of threat-related compared with neutral words [20,43]. Traditionally, slower colour-naming for threat words has been considered to reflect an “attentional capture” by threat words: the threat content of the stimulus is processed faster, or receives more attentional resources, than the non-threatening colour component (e.g., [42]). Recent work, however, suggests that the emotional Stroop task not only provides a measure of attentional capture, but also reflects response slowing as a result of subtle cognitive freezing induced by the mildly threatening content of stimuli [20,22,23]. This slowing effect may be detected using other classes of emotional stimuli like faces [38].

Recently, a modified version of the “emotional Stroop” task was developed with chimpanzees [44]. In this study, chimpanzees were trained on an adapted version of the emotional Stroop task. Chimpanzees initially learned a colour discrimination task in which two coloured square frames would appear on the screen. Chimpanzees were rewarded for touching the blue frame, but not the yellow frame. In the first experiment, blue and yellow abstract shapes appeared in the frames and chimpanzees were slower and less accurate in responding to blue frames when the abstract shape was yellow than when it was blue, indicating classic Stroop interference on these incongruent trials. In a second experiment, colour photographs of humans (veterinarian, caretaker and stranger) were presented in the colour frames. Chimpanzees were slower to touch the blue frame target when a picture of the veterinarian was present than on trials in which pictures of a stranger or caretaker were present, and this effect was stronger in animals who had more recently been anaesthetized by the veterinarian. The increased task interference and slowing of correct responses on trials with the picture of the veterinarian were interpreted to be a result of negative emotional valence associated with veterinary stimuli. Training on the emotional Stroop task ranged between 900 trials over nine sessions for the fastest animal to reach criterion to 6700 trials over 67 sessions for the slowest animal. Of the 16 animals available at the start of the study, seven completed training and testing on the variant of the emotional Stroop task [45]. A current issue with cognitive tasks is the extent to which the populations we work with are self-selecting, and so tasks that are easiest to perform may be most promising for wider use. We propose here a simpler “response slowing” task [24] which may measure the same processes that were measured by Allritz *et al.* [44] but due to the simpler learning criteria may be less susceptible to attrition of subjects.

Spatial cueing and visual probe tasks in humans have also revealed response slowing to mildly threatening distractor stimuli, such as angry faces, again reflecting what may be a subtle form of freezing [21,24]. For example, in their central cueing task Mogg *et al*. [24] presented participants with a grey square on a computer screen followed by a face (angry, happy or neutral) in the centre of the square. A target (up or down arrow) then replaced the square and the participant responded by pressing one of two buttons to indicate whether the arrow pointed upwards or downwards. Angry faces significantly slowed responses in individuals who were high in both trait and state anxiety (but not those who reported low anxiety levels), revealing an important association between threat-related response slowing and individual differences in emotion state.

In another study, Fox *et al.* [21] presented half of the participants with negative and highly arousing pictures (e.g., disaster scenes), increasing their state anxiety prior to their performing a spatial cueing task. Participants in an anxious state were faster than those in a non-anxious state to respond on trials when neutral or positive emotional content was presented (possibly reflecting some anxiety-related arousal [46]). Despite this general overall speeding of responses in the state anxious group, these participants were significantly slower to respond on trials when angry faces were presented than when either happy or neutral faces were presented, suggesting “*a subtle cognitive form of the freeze response found in animals*” [21] (p. 698).

Methods that tap into subtle cognitive forms of processes that are associated with dysregulated emotion and negative affect may prove valuable for assessing animal welfare. Studies using primate models of human psychopathology have provided data on similarities between human and other primate species in the neural substrates underlying cognition-emotion interactions (e.g., [15,18,47,48]). These data suggest that it is appropriate to adapt cognitive tasks that tap into human emotion to explore similar processes in primates. In addition, animal welfare research has seen a recent surge of interest in developing cognitive measures of animal emotion. Widespread uptake of “cognitive bias” methods, as first developed by Harding *et al*. [1], has resulted in a body of data and new methodological advances [9,20,22,24,29,30,44,49,50] to which, we propose, the response slowing approach would make a valuable addition. In line with the attention bias approach [8,11], tasks assessing emotion evaluation and response slowing to mild threat should also be relatively simple to measure and interpret. Due to their reliance on biologically relevant stimuli, response slowing tasks should require less initial training than do judgement bias tasks. Furthermore, performance on such tasks may be more readily interpreted in terms of underlying emotional responses, depending on the intrinsic biological relevance of the stimuli employed [14,15]. Tasks that tap into subtle pre-cursor components of the freeze response may therefore be important to the development of welfare measures of individual coping ability and vulnerability to stress, particularly as they are straightforward in their implementation and interpretation and may not be encumbered by the ethical concerns that some other fear-eliciting studies may face.

Here, we present a task to measure response slowing to mild threat in captive rhesus macaques. We included an emotion manipulation, as is common in the cognitive bias approach, in order to assess the extent to which putatively negative shifts in emotion state would influence response slowing to threat. In light of our findings, we discuss how methods that tap pre-cursor processes to fear, such as the subtle cognitive freeze response to mild threat, may provide a valuable assessment tool for animal emotions.

## 2. Methods

### 2.1. Participants and Housing

Ten sexually mature male rhesus macaques (*Macaca mulatta*) housed at the Sabana Seca Field Station, Caribbean Primate Research Center (CPRC), Puerto Rico, took part in the study (Mean age: 7.39 years; range: 3.60–24.7 years old). All monkeys were captive born and housed in an outdoor, covered enclosure in single quarantine caging in accordance with United States federal regulations. The monkeys had access to water *ad libitum* in the home cage and were provisioned with 20% protein, 5% fat, 10% fibre commercial dry primate diet (Diet 8773, Teklad NIB primate diet modified, Harlan Teklad, Madison, WI, USA) supplemented with fruit during morning and afternoon feeding rounds. The study conformed to the University of Puerto Rico’s Institutional Animal Care and Use Committee (IACUC) Guidelines (Protocol approval: A1850106) and was passed by the Ethics Committee of Roehampton University. Monkeys had access to water *ad libitum* throughout training and testing, and were fed a portion of the day’s food ration at the end of each daily testing session, regardless of performance. Monkeys were tested sequentially in the same order each day in a testing area adjacent to the housing where they had no visual access to other monkeys, and very limited auditory access. Testing was conducted opportunistically with animals while they were in quarantine. Details of training are given in the Supplementary Materials. In line with best 3Rs practice [51,52], monkeys were moved to pair-housing in larger, purpose-built, floor-to-ceiling enclosures for welfare purposes at the end of the quarantine period.

### 2.2. Stimuli and Apparatus

Stimuli were presented on a 15” Protouch Aspect TS17LBRAI001 touch-sensitive LCD monitor, connected to a Toshiba Satellite Pro A60 laptop computer running EPrime experimenter-generator software [53]. A stimulus consisting of a grey rectangular frame measuring 154 mm × 164 mm was composed for training and as an experimental control during testing (Figure 1a). Distractor content for the stimulus comprised 20 colour photographs of 10 male monkeys (“stimulus monkeys”). Photographs were taken by Emily J. Bethell. of males housed at the facility who were unlikely to be known to the participant monkeys. For each stimulus monkey, there was one frontal face picture with neutral expression (hereafter “direct gaze”), and one profile view face picture with neutral expression (“averted gaze”: Figure 1b). Images were trimmed so that only the monkey’s head was visible, and superimposed onto the grey stimulus used for training and control trials. All stimuli subtended 14.7 × 15.7 degrees of visual angle when presented centrally on a computer monitor at a 60 cm viewing distance. Direct/averted gaze face pairs for each stimulus monkey were matched for luminosity (*Ly)* and contrast energy (*C*) using Adobe Photoshop 7.

Monkeys were rewarded for touching the stimulus on each trial with automatic delivery of a Noyes 190 mg food pellet triggered by a pellet dispenser (Biomed Associates Pedestal 45 mg mount dispenser, ENV-203) via a tube to a pellet tray in front of the monitor. A secondary reinforcing tone was played via two speakers behind the apparatus. Primary and secondary reinforcers were delivered on a 100% fixed reinforcement ratio during testing. All responses were recorded automatically by the computer.

### 2.3. Procedure

Monkeys were initially trained to touch the grey square control stimulus at each of three locations on the screen. A trial lasted for 60 s or until the monkey touched the stimulus, whichever occurred sooner. During testing, monkeys continued to be rewarded for touching the control stimulus, but, on some trials, distractor face content was present (Figure 2). These trials were also rewarded. At stimulus offset, a plain black screen was shown until the onset of the next trial. A variable inter-trial interval, with a minimum duration 8080 ms, allowed monkeys time to collect and eat pellets in between trials.

**Figure 1 behavsci-06-00002-f001:**
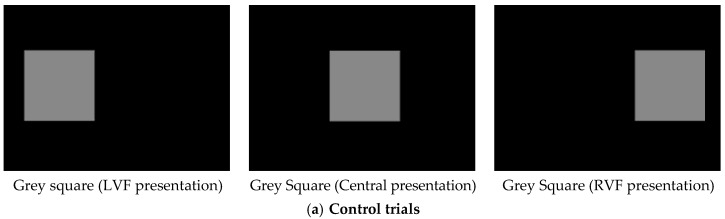
Examples of stimuli as they appeared at the three screen locations for (**a**) Control trials; and (**b**) Experimental trials.

**Figure 2 behavsci-06-00002-f002:**
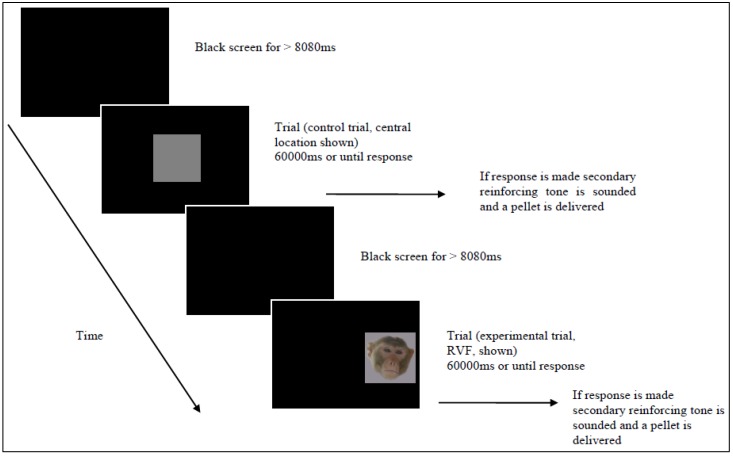
Example of the experimental procedure.

A testing session consisted of three blocks. Block 1 consisted of three practice trials, during which the control stimulus was presented once at each of the three locations. Monkeys were required to touch the stimulus on all three trials in order to move onto experimental trials (Block 2). Block 2 consisted of 75 trials: 60 experimental trials interspersed with 15 control trials. Presentations of the 20 experimental stimuli (10 direct gaze faces and 10 averted gaze faces) were balanced so that each experimental stimulus was presented three times, once at each of the three screen locations. The control stimulus was presented five times at each location.

The testing schedule is shown in Figure 3. Data were collected from monkeys on two days during a baseline condition (“baseline”) and on the two days immediately following a putatively stressful veterinary examination (“stress”). We “piggy-backed” this study on the pre-existing three-monthly veterinary examination during which monkeys are restrained and anaesthetised for physical examination (details given in Supplementary Materials). Order of testing was counterbalanced so that five monkeys were tested first in the “baseline” condition, and five monkeys were tested first in the “stress” condition. “BL first” monkeys received four days of contingency training concurrent with enhanced enrichment in the home enclosure (days −4 to −1). On day 0 and day 1, they had two daily testing sessions (“baseline”). They then continued to take part in daily maintenance sessions until day 9, when health-checks were conducted for all monkeys in the study. The “stress first” group started their four days of contingency training on day 5 and underwent the veterinary health check on day 9. On days 10 and 11, all monkeys underwent testing in the “stress” condition, after which they received extra cage enrichments. The “stress first” monkeys then took part in daily maintenance sessions until days 19 and 20 when they underwent testing in the “baseline” condition.

**Figure 3 behavsci-06-00002-f003:**
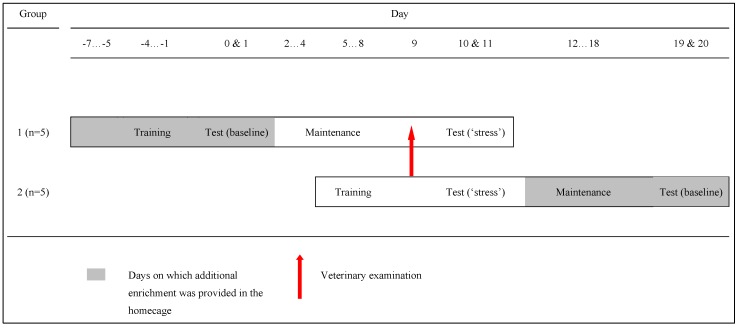
Training and testing schedule showing counterbalanced order of testing.

### 2.4. Data Treatment and Statistical Analysis

Criterion for inclusion of each monkey’s data in the analyses was ≥12/15 (80%) responses on control trials on at least one daily testing session in each testing condition. In line with treatment of reaction time data in human studies (e.g., [21,24,38,39]) latency data were trimmed so that non-responses and responses faster than 400ms were discarded (Supplementary Materials). Due the large variability inherent in reaction time data (which may be enhanced in animals, who have not been instructed to work as quickly as possible as is the case for humans) data were normalized (e.g., [13]) using a Log_10_ transformation, and means calculated for analysis by repeated measures analysis of variance (RMAnova), again in line with, and to allow cross-reference with, human studies (e.g., [6,7,12,13,20,21,22,23,24,54]). In order to control for the well-documented and generalized effect of arousal on latency to respond (*i.e.,* independent of stimulus valence) we followed the human literature in which participants are used as their own control by calculating difference scores for responses to emotional versus neutral stimuli (e.g., [23,24,55,56,57]). To allow for the large degree of inter-individual variation in animal reaction time data enhanced by our long trial duration of 60 seconds (and following empirical data from humans showing that reaction time ratio scores are more robust to floor effects [58,59] and that ratio indices show convergent validity with difference scores [60]) we calculated difference ratio scores separately for direct gaze and averted gaze trials as follows:
(1)
[Ratio = (mean log10 latency experimental trial/mean log10 latency control trial)]

One-sample Kolmogorov-Smirnov tests revealed transformed data did not differ significantly from a normal distribution; therefore, parametric tests were used throughout. For the paired *t*-tests, a Levene’s test of equality of variance was also conducted. The results of this test were all non-significant. All descriptive data are reported as mean ± 1SE. Significant findings are presented in figures. Full details of non-significant findings are given in the Supplementary Materials.

## 3. Results

Training data are given in Supplementary Materials. Seven monkeys reached criterion for inclusion in the study: six monkeys reached criterion on all four of their testing sessions (=24 sessions in total); one monkey reached criterion only on one day in either condition (=2 sessions). Three monkeys failed to reach criterion on at least one day in either treatment (Supplementary Materials). The seven monkeys who reached criterion responded faster than 400 ms on 173 trials, and did not respond at all on 44 trials (there was no difference between “baseline” and “stress” conditions in number of trials faster than 440ms, or in number of non-responses: Supplementary Materials). These trials were removed, resulting in 1733 trials from seven monkeys being entered into the analysis.

### 3.1. Emotion State and Emotional Distractor Interact to Impair Performance

A 2 × 2 × 3 RMAnova was performed to determine whether the presence of emotional distractors impaired performance on the task. Data were ratio scores, with within-subjects factors of condition (“baseline” *versus* “stress”), trial type (direct gaze and averted gaze) and visual field (left, central and right). There was a significant main effect of condition (F_1,6_ = 11.749, *p* = 0.01): ratio scores >1 in the “stress” condition revealed slowing of responses on experimental trials relative to control trials, while ratio scores <1 at “baseline” revealed speeding of responses on experimental trials relative to control trials. There was a significant interaction between testing condition and trial type (F_1,6_ = 6.68, *p* = 0.04: Figure 4). There were no other main effects (trial: F_1,6_ = 1.786, *p* = 0.230; visual field: F_2,12_ = 0.249, *p* = 0.784). To examine the two-way interaction in more detail, data were collapsed for visual field, and planned pairwise comparisons were conducted. A paired samples *t*-test, with within-subjects factor testing condition (“baseline” *versus* “stress”) revealed a significant difference in the ratio scores for direct-gaze trials between the two conditions (t_6_ = 3.41, *p* = 0.01). There was no effect of condition for averted gaze trials (t_6_ = 1.82, *p* = 0.12). One-sample *t*-tests (test value = 1) were conducted for the direct-gaze trials to examine whether responses were significantly faster or slower on these experimental trials than control trials. These revealed responses on direct gaze trials did not differ significantly from speed to respond on control trials in either condition: both *t* > 1.70, both *p* > 0.13. Finally, we calculated residual values for latency to respond on direct gaze trials at baseline plotted against latency to respond on direct gaze in the stress condition. We ran a bivariate correlation to test whether baseline response to direct-gaze trials predicted magnitude of response slowing following the stressor. There was no significant relationship (*r* = 0.00, *n* = 7, *p* = 1.00).

**Figure 4 behavsci-06-00002-f004:**
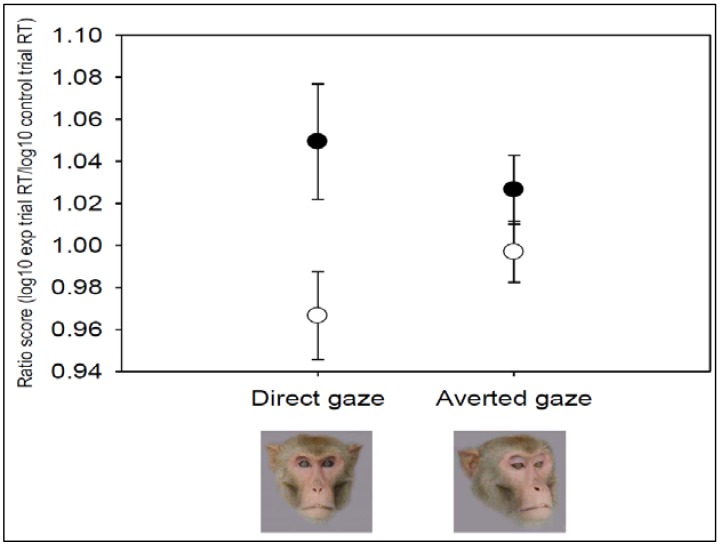
Ratio scores for latency to respond on experimental trials with direct gaze and averted gaze emotional distractor content. ● = “stress”; ○ = “baseline”.

### 3.2. Additional Tests

*Post hoc* analyses were conducted to check for possible priming effects of previous trial shown, and to test possible effects of stress-related arousal.

#### 3.2.1. Previous Stimulus Seen Does Not Affect Latency to Respond in Current Trials

A 3 × 2 × 3 RMAnova was performed to examine whether the stimulus presented on the previous trial influenced latency to respond on the current trial. Data were log_10_ transformed latencies, with within-subjects factors current trial (control, direct gaze, averted gaze), testing condition (“baseline” *versus* “stress”) and previous trial (control, direct gaze, averted gaze). There was no main effect of previous trial on response latency on the current trial (F_2,8_ = 2.435, *p* = 0.149), nor were there any interactions (all Ps > 0.5).

#### 3.2.2. Stress-Related Arousal Speeds Responses on the Basic Operant Tasks

A 2 × 3 RMAnova was run to test the hypothesis that stress-related arousal enhances speed to respond to non-emotional control stimuli. Data were Log_10_ latencies for control trials, with within subject factors of testing condition (“baseline” *versus* “stress”) and screen location (left, central and right). There was a main effect of testing condition (F_1,6_ = 17.611, *p* = 0.006): monkeys were faster to touch control stimuli in the “stress” condition (mean RT: 4.7 s ± 1.7) compared to “baseline” (mean RT: 9.6 s ± 2.5). There was no main effect of screen location (F_2,12_ = 0.307, *p* = 0.741), and no interaction of testing condition x screen location (F_2,12_ = 1.561, *p* = 0.250).

#### 3.2.3. Food Motivation Does Not Differ between Conditions

We ran a 1 × 3 RMAnova to examine whether motivation to work, measured as proportion of pellets eaten, varied with condition. Data were proportion of pellets consumed with within-subjects factor of training/testing condition (training, “baseline” and “stress”). The proportion of pellets eaten did not differ significantly between training and the two testing conditions (F_2,12_ = 2.204, *p* = 0.203). Planned t-tests revealed there was no significant difference in pellet consumption between the testing conditions (“baseline” *versus* “stress”: t_6_ = 1.55, *p* = 0.17), nor between either of the testing conditions and training (“baseline” *versus* training: t_6_ = 1.32, *p* = 0.24; “stress” *versus* training: t_6_ = 1.44, *p* = 0.20). All monkeys consumed the full daily food ration while in the laboratory after each training and testing session.

## 4. Discussion

We developed a response slowing paradigm for use with non-human primates, based on methods used to assess the role of emotion in attention for threat in humans. When they had previously experienced a stressor (veterinary examination) monkeys were slower to respond to mildly threatening (direct gaze) stimuli, relative to their speed to respond during a baseline period, when generalised arousal effects were controlled for. There was no evidence for response slowing to non-threatening (averted gaze) social stimuli. This demonstrates that a subtle cognitive form of a freeze response to mild threat may provide an indicator of negative affect in a non-human animal. This study has implications for our understanding of animal emotion and attention and for the development of more ethical methods for assessing fear and anxiety—and consequently welfare—in animal research models.

The finding of slower responses to mildly threatening stimuli following the environmental stressor is in line with data from humans showing response slowing to threat faces in anxiety, which has been attributed to a subtle cognitive form of the freeze response [21,24]. Our finding of response slowing to threat following a stressor, but not during a presumably less stressful baseline period, would support an interpretation in terms of a subtle freeze response as a precursor to negative affective states such as anxiety or fear (as identified in humans [21,24]). This interpretation is supported by evidence, firstly, from primate models of human psychopathology which report enhanced bodily freeze responses to threat among primates subjected to early life stressful events such as separation from the mother during infancy [16,18,19]. Further support is provided by studies demonstrating homologous brain substrates involved in processing threat-related stimuli across species including humans [17,47,48].

Although monkeys showed response slowing to direct-gaze stimuli in the stress compared to the baseline condition, within each condition, there was no significant difference in speed to respond on experimental trials compared to control trials. This suggests that the cognitive freeze effect underlying response slowing was, indeed, subtle. The response slowing paradigm we present here may therefore be most appropriate for identifying relatively large shifts in affect (we would argue that veterinary inspections are highly stressful events for captive animals), and may lack sensitivity for detecting less extreme shifts in emotion. Indeed, researchers using animal models of human psychopathology often use an “extreme groups” approach [18,37], only selecting for study those animals who show the most extreme patterns of behavioural response. The method presented here is sensitive to more subtle indicators of the fear response than full bodily freeze, and offers a potential refinement to methods traditionally used with animal models of human psychopathology.

There was no effect of the presence of non-threatening averted-gaze distractor faces on monkeys’ arousal-corrected speed to respond in either condition. Ratio scores for averted-gaze faces were equivalent between the baseline and stress conditions. This finding indicates that it is the emotional relevance of the distractor content that influenced response speed, rather than the presence of social information *per se*. Direct gaze is a signal of dominance in rhesus macaques, while averted gaze signals submission [34,61]. It is therefore likely that the significant difference in ratio scores for direct-gaze faces between the baseline and stress conditions is associated with the socio-emotional significance of those faces to adult male macaques, mediated by their emotion state. This adds to the growing body of studies that have effectively used emotional face stimuli to test socio-cognitive processes in non-human primates, [8,62,63,64,65,66,67,68,69,70].

We found no evidence for priming effects of the previous trial. This suggests that the relatively long inter-trial-interval (>8 s) was sufficient to avoid carry-over effects for the seven monkeys who completed the study. However, it is possible that the inter-trial interval was not sufficient for the three monkeys who failed to reach criterion during testing and were subsequently removed from the analysis. For these monkeys, inhibitory carry-over effects of previous trials may have caused them to stop responding on control trials and to disengage from the task altogether [22,23]. It would be interesting for future studies to consider which characteristics of animals and stimuli influence ability to complete such tasks, or even to begin training. Likely factors include genotype [67], sex [62,63], temperament and dominance [62,63,64,70], as well as recent exposure to threat [8,44], social competition [71] or mating cues [65]. An alternative explanation for the three monkeys failing to reach training criteria is that they may have had a strong attention bias for the direct gaze faces, which distracted them from the ongoing task. Our previous work shows that adult male macaques exhibit rapid and sustained attention for threatening faces, with a large degree of individual variation in the extent to which face stimuli capture and hold attention [8]. It is possible therefore that the extent of attentional capture led to failure to continue on the basic task.

There were several factors that we aimed to control for in the study design. First, we controlled for arousal related changes in response speed between the two conditions by including control trials in which responses to grey rectangular stimuli were required. Monkeys were faster to respond on control trials in the stress condition than at baseline. This is in line with well-established relations in humans between anxiety, arousal and response speed e.g., [28,72]. We were able to account for this effect of arousal-related speeding of response in the stress condition by calculating ratio scores for analysis. The use of difference scores is standard practice in the human literature and their use allows each participant to act as their own control [24,55,56,57], a potentially valuable approach especially when working with groups with longer response times and thus greater inter-individual variability in response latencies. Second, we controlled for possible effects of lateralization of emotion processing by presenting stimuli centrally, and at left and right peripheral locations, on the screen. While we included visual field in our analysis to explore possible hemispheric bias effects, it is perhaps unsurprising that we found none; the monkeys in our study were unrestrained, and there was no requirement to focus on a central cue prior to stimulus onset as would be required to investigate visual field effects e.g., [67,70]. Finally, we controlled for potential order of testing effects by testing half of the monkeys first in the stress condition and half of the monkeys first in the baseline condition.

The finding of response slowing to conspecific faces with direct gaze suggests that we do not need to use extreme fear-inducing stimuli, such as a stranger walking into the room [19], to detect early indicators of dysregulated fear response in macaques. Instead, use of milder stimuli, similar to those used in human cognitive research (*i.e.*, pictures of faces [21,24,39]) may provide a more ethical and sensitive approach. We are currently exploring attention biases for face stimuli along a gradient of emotionality to identify those stimuli that may be most readily categorised as “neutral”, “mildly threatening”, or “threatening” to rhesus macaques. A future extension of this work is to investigate the utility of using positive stimuli for assessing animal welfare. For example, here we have identified response slowing to direct gaze faces as a subtle cognitive form of freezing to mild threat. Studies with humans have found that novel and positive stimuli (e.g., smiling faces) may also elicit a strong orienting response and behavioural inhibition associated with slowed response on an ongoing task [24,55,57]. Mogg *et al*. [24], for example, found that both high and low anxious people were slower to respond on trials on which smiling faces were presented, while only high anxious people were slower to respond on trials in which angry faces were shown. Differential response slowing to differently valenced stimuli may therefore provide power to discriminate between different emotions. From a practical perspective, inclusion of positive stimuli may also improve retention of subjects.

From a theoretical perspective, this intersection of animal and human work—we use monkey participants with a paradigm that is typically used with humans to study cognitive aspects of emotion processing—may inform our understanding of the evolution of fear and defense behaviours in humans and other animals [14,15,17,25,28,31]. As well as furthering our understanding of the adaptive function of fear and defense behaviours, findings from tasks such as these may enhance our understanding of the extent to which orienting responses occurring after the detection of threat may be associated with human stress and psychopathology. For example, a heightened freeze response to mild threat is associated with dysregulated fear and impaired wellbeing, as is seen in post-traumatic stress disorder [30].

From an ethical perspective, there are welfare benefits that could arise from devising methods that are sensitive to pre-cursor processes to what might otherwise be highly negative emotional responses to potential stressors. The use of response slowing methods to assess levels of subtle freezing to mild threat, without having to evoke a full fear response to highly threatening stimuli (e.g., HIP: [16,19,37]), provides a more ethical approach to the assessment of animal emotions related to fear. These methods also provide detail on the mechanisms underlying dysregulated fear and impaired wellbeing in captive animals.

From an applied perspective, the use of stimuli that tap into early and automatic attentional processes may prove useful for the development of veterinary tools that can be used to quickly and efficiently identify animals that have dysregulated emotions and are therefore more vulnerable to developing affective disorders. We argue attention bias and response slowing tasks should require less training than the standard judgement bias task [4] (see also [2,11]) and the newly adapted emotional Stroop task [44] and therefore be quicker to implement. We acknowledge one primate judgement bias study [73] which was conducted with three chimpanzees who required 5–10 training sessions (comparable to the 3–16 sessions reported here), and that there are likely to be species and individual differences in how well animals learn the tasks. Of the 11 chimpanzees who successfully learned to use the touchscreen in [44], two animals subsequently failed to learn the initial categorization task after 80 sessions and were dropped from the study [45]. Other research groups also report great variation in primates’ ability to learn the initial discrimination required to perform cognitive bias tasks [74], suggesting a need for more adaptable methods with more simple learning criteria. Quick and easy tasks may be especially useful for working with “at risk” groups such as animals who have experienced early life stress [75], trauma [76], or are subjected to potentially stressful or invasive research procedures with cumulative suffering implications [77,78]. Furthermore, these tools may be useful for identifying the types of stimuli that individual animals find threatening. For example, Allritz *et al*. [44] found response slowing to veterinary stimuli, but not to caretaker or stranger stimuli, suggesting the veterinarian is a threatening stimulus to captive chimpanzees, whose saliency diminishes with time since last capture for anaesthetisation. For application for welfare assessment purposes, it is important to distinguish the benefits of these measures for (a) informing our understanding of species-level general trends (e.g., response slowing to threat signifying subtle freezing, which may be enhanced in some species compared to others) and (b) their potential application for developing individual profiles (*i.e.*, how changes in response for an individual animal might signify changes in affect over time). It is possible that increases or decreases in an individual’s subtle freeze response to mildly threatening stimuli could provide an early indicator of the onset of dysregulated emotion, providing a diagnostic tool for identifying “at risk” individuals. With greater understanding of the processes underlying animal cognition and emotion, we can design methods that use relevant stimuli, require fewer trials, are quicker and cheaper to conduct than some existing methods, and have the power to distinguish positive and negative shifts in affect.

## 5. Conclusions

In summary, the response slowing paradigm we present here has a number of advantages over traditional research methods. Traditional methods that use overt fear and freeze responses are unethical as animal welfare measures. Existing welfare methods such as judgement biases and the recently presented emotional Stroop paradigm developed with chimpanzees [44] require require long periods of training and may not be learned by all animals. We encourage researchers to consider the response slowing paradigm as a complementary method that may further our understanding of emotion-evaluation, defensive behavior and dysregulation of emotions like fear in animals, and extend the methods available to assess and improve animal welfare in captivity.

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
