# Peer review of "Emotion Evaluation and Response Slowing in a Non-Human Primate: New Directions for Cognitive Bias Measures of Animal Emotion?"

_behavsci, 2016, doi:10.3390/bs6010002_

Round 1
Reviewer 1 Report
This is an interesting and timely paper which reports a test of emotional response in macaques using a paradigm comparable to that recently reported with chimpanzees. I think this paper would be of interest to many and my only comments concern the presentation of the methods. In particular, I would like to have seen more detail throughout and I provide comments specifying what additional information would be useful.
I also note that one of the authors’ key motivations for using these new methods is that they are quicker to implement (less training and testing time). Given this, I was surprised that a recent study with chimpanzees (Bateson & Nettle, 2015) was not cited, as this study tested a new method for testing primates’ emotions that a) did not require the animals to be touchscreen trained and b) implemented a fast training paradigm. I would recommend that this paper be cited: Bateson M & Nettle D 2015 Development of a cognitive bias methodology for measuring low mood in chimpanzees. PeerJ 3:e998 https://doi.org/10.7717/peerj.998
Page 5 (2.1 Participants and Housing) – please provide more detailed information about the monkeys’ housing. For example, were they housed singly, in pairs, or in groups? What size were their enclosures? Was the meal they received at the end of testing their only meal of the day? Was it withheld if they did not participate in the testing? Were they on a reduced-weight diet? What were they fed? Could the monkeys see the other subjects participating in the study? Where they tested simultaneously or sequentially?
Page 6 (2.2. Stimuli and Apparatus) – Were the stimulus monkey photographs also of rhesus macaques? Did you take these photographs or were they stock images? (if the latter, please provide credits for the photographs).
Page 6 (2.3 Procedure) – How long did this training for the 'simply attentional task' take? How many trials and over what period of time? Equally, how long did testing take and over what period of time? Was there a range for the different monkeys?
Page 7 (Fig 2) – so the monkeys were rewarded for touching the face stimuli 100% of the time as they were the grey squares, correct? What if the monkeys made no response? And how often did that happen?
Page 8 (Fig 3) – how come Fig 3 shows total n=12 when in your methods you say you only tested 10 monkeys? This figure also mentions a “health check”. Given that this is the key part of your methodological design, please provide more details on this in your methods. Were the monkeys anaesthetized for this? How regularly did such health checks occur? Were they removed from their cage for these health checks? How long did the health check last?
Reviewer 2 Report
Overall this is an interesting paper which explores a novel area of research relevant to both evolutionary theory and welfare applications. Whilst it is mostly clearly written, I have a number of recommendations for the authors, primarily relating to clarifying a number of points for the reader. The most major issue is with the analysis - the authors need to be clear on why they used a ratio score instead of individual measures for experimental/control trials, and why they calculated mean scores instead of including all data points in the analysis. If the latter cannot be well justified, the data should be re-analysed without mean scores, using all data points available.
Abstract:
· The first sentence doesn’t read clearly, I’d suggest breaking it up.
Introduction:
· Overall the introduction is clearly written and gives the reader a good understanding of both the background literature and recent work/theories that have led to this study. The links to welfare are interesting and potentially important; it might be worth emphasizing this more clearly early in the introduction, i.e. how does this work help welfare assessment.
· Page 3 line 24: for readers less familiar with dominance/threat behaviour in macaques, it would be worth emphasizing the meaning/importance of direct eye gaze/stare in the introduction to help them to understand the rationale behind the experimental procedure (refer to line 22, page 11).
Methods
· Authors should include an ethics statement, including the project identification code, date of approval and name of the ethics committee, in accordance with author guidelines for this journal
· Page 5, 2.1: are all subjects adults? Page 11, line 24 mentions adult male macaques, but whether all subjects are adults is not clearly stated in the methods. Can the authors please clarify this.
· Page 5, 2.1: the authors state that the monkeys were housed in quarantine during testing – does this mean that they were socially isolated? The authors should state whether animals were socially housed and whether they had been socialised previously, given the social nature of the experimental stimuli.
· Page 5, 2.1: If the monkeys were socially housed, do the authors have rank data on the animals? (see notes on discussion).
· Page 8, 2.3: in the discussion the authors explain their use of ratio scores, instead of using separate scores for experimental/control conditions, as a way to account for speeding of response in the stress condition (page 12, line 1). This should be made clear in the methods along with the equation.
Results
· Spell out RMAnova as repeated measures Anova for first use.
· Why have the authors collapsed scores across trials? This really needs to be explained. If running a repeated measures analysis, you should be able to account for multiple measures per monkey per condition in the analysis, without calculating mean scores first. If there was a specific reason for calculating the mean score prior to running the analysis, this should be clearly justified.
· Page 9, line 5; line 15; page 10, line 3: can the authors please provide all null results in a table for the reader to refer to. Supplementary material would suffice.
· Provide the t value on page 9 line 15 to 2 dp consistent with other results.
Discussion
· Overall the discussion is clearly written and explores the potential impacts of this research well.
· Page 11, lines 14 - 17: another angle to consider here in regards to welfare and assessing individual response (something that you touch on the intro, and would be worth bringing up here), is that individual response may be affected by differences in baseline levels of anxiety or stress – i.e. monkeys higher in anxious trait or stress-related behaviours may exhibit stronger responses in the stress condition.
· Furthermore, given that the authors mention eye gaze being an indication of dominance, it is worth considering whether rank of test subjects could also affect their response to direct vs. averted gaze. One would expect that lower ranking individuals would find direct gaze more threatening than higher ranking individuals. This could account for the ‘large degree of individual variation’ in response to faces that the authors mention (page 11, line 35). I suggest that the authors either include this as a predictor variable if the data is available, or add it to the discussion for future consideration.
References
· Reference 55 needs formatting correctly.
Typos/formatting issues:
· Page 2, line 20: consider using spelling’ judgement’ instead of ‘judgment’
· Page 3, line 1: space required between ‘processthe’
· Page 3, line 16: space required between ‘2seconds’ and ‘50seconds’; page 10, line 9: ‘4.7seconds’, ‘9.6seconds’; page 11, line 28: ‘>8seconds’
· Page 3, line 22: be consistent in use of either ‘seconds’ or ‘s’
· Page 10 lines 9-10: space out + and -
Round 2
Reviewer 1 Report
The authors have responded to all my comments and the article is now much more clear.